# Changes in the Fatty Acid Profile in Erythrocytes in High-Level Endurance Runners during a Sports Season

**DOI:** 10.3390/nu16121895

**Published:** 2024-06-16

**Authors:** Francisco Javier Alves Vas, Fco. Javier Grijota Pérez, Víctor Toro-Román, Ignacio Bartolomé Sánchez, Marcos Maynar Mariño, Gema Barrientos Vicho

**Affiliations:** 1Faculty of Education, University Pontificia of Salamanca, 37007 Salamanca, Spain; fjalvesva@upsa.es (F.J.A.V.); ibartomesa@upsa.es (I.B.S.); gbarrientosvi@upsa.es (G.B.V.); 2Sport Science Faculty, Department of Physiology, University of Extremadura, 10003 Caceres, Spain; mmaynar@unex.es; 3Faculty of Health Sciences, Isabel I University, 09003 Burgos, Spain; 4Research Group in Technology Applied to High Performance and Health, Department of Health Sciences, Universitat Pompeu Fabra, TecnoCampus, 08302 Mataró, Spain

**Keywords:** endurance training, fatty acids, arachidonic acid, docosahexaenoic acid, omega-3

## Abstract

Fatty acids (FAs) are an essential component of the erythrocyte membrane, and nutrition and physical exercise are two variables that affect their structure and function. The aim of this study was to evaluate the erythrocyte profile in a group of high-level endurance runners, as well as the changes in different FAs, throughout a sports season in relation to the training performed. A total of 21 high-level male endurance runners (23 ± 4 years; height: 1.76 ± 0.05) were evaluated at four different times throughout a sports season. The athletes had at least 5 years of previous experience and participated in national and international competitions. The determination of the different FAs was carried out by gas chromatography. The runners exhibited low concentrations of docosahexaenoic acid (DHA) and omega-3 index (IND ω-3), as well as high values of stearic acid (SA), palmitic acid (PA), and arachidonic acid (AA), compared to the values of reference throughout the study. In conclusion, training modifies the erythrocyte FA profile in high-level endurance runners, reducing the concentrations of polyunsaturated fatty acids (PUFAs) such as DHA and AA and increasing the concentrations of saturated fatty acids (SFAs) such as SA and the PA. High-level endurance runners should pay special attention to the intake of PUFAs ω-3 in their diet or consider supplementation during training periods to avoid deficiency.

## 1. Introduction

Exercise intensity, duration, distribution throughout a season, and recovery periods are essential variables in prescribing training for endurance runners [1]. Suitable nutritional intake is fundamental for recovery processes, adaptation, and optimizing performance in competition [2]. The roles of carbohydrates and proteins in endurance athletes have been extensively documented, continuing to be a subject of interest for performance optimization and recovery [3,4]. However, lipid intake receives less consideration by endurance athletes [5].

Fatty acids (FAs) play a crucial role in the athletes’ organisms, serving as energy stores and precursors to certain hormones, participating in the immune system, and being a structural component of cell membranes [6,7]. It is known that the composition of FAs ingested through the diet and their metabolism in the body correlates with the FAs that are part of different structures such as erythrocyte membranes, playing a crucial role in their function [8,9].

Special interest has been given to polyunsaturated fatty acids (PUFAs) in athletes: essential fatty acids that must be introduced through the diet [10]. PUFAs nutritionally important for human health are members of the so-called omega-3 (ω-3) and omega-6 (ω-6) families. Within the ω-6 family, arachidonic acid (AA) forms proinflammatory eicosanoids, especially prostaglandins, thromboxanes (potent vasoconstrictor), and leukotrienes that, in excessive amounts, contribute to thrombus formation, atheromas, and inflammatory disorders [11]. In relation to the ω-3 PUFA family, docosahexaenoic acid (DHA) and eicosapentaenoic acid (EPA) stand out for their beneficial effects. ω-3 intake increases prostacyclin concentrations, which have vasodilatory and platelet aggregation inhibitory properties, balancing the activity of proinflammatory eicosanoids [12]. Another benefit observed in athletes is an increase in cell membrane fluidity, improving oxygen diffusion to tissues, as fluidity is determined by the presence of PUFAs located on both sides of lipid bilayers [13]. Peoples et al. [14] found a reduction in myocardial oxygen demand during exercise in a group of well-trained cyclists when their ω-3 concentrations increased. Besides, ω-3 participates in muscle protein synthesis, promoting recovery in athletes [15], and may enhance immune system function [16]. In erythrocytes, values of ω-3 below 8% may have negative effects on their integrity, associated with a higher risk of cardiovascular issues [17]. Deficits in this rate have been documented in various groups of athletes [18,19].

Regular training induces changes and adaptations in hematological and biochemical parameters that can affect erythrocyte cell membranes [20]. Exercise generates reactive oxygen species (ROS) and free radicals that have a disruptive effect on cells, requiring repair [21]. Erythrocyte membranes contain a high amount of PUFAs, and together with high amounts of oxygen and heme iron, making them susceptible to lipid peroxidation (LP) and potential muscle damage [22]. The peroxidation process involves the oxidation of a FA, converting it into a radical that can oxidize adjacent molecules [23]. The erythrocyte lifespan is around 120 days, making them a suitable matrix for measurement, reflecting their composition and long-term adaptations [24].

The type of exercise is another factor that modifies the FA profile in different tissues [25], as it requires greater mobilization and oxidation of FAs among tissues as an energy substrate [26]. Moderate-intensity exercise has been reported to increase prostacyclins and decrease thromboxanes, favoring vasodilation and improving the body’s immune and anti-inflammatory response [27]. However, high-intensity training increases the generation of superoxide radicals in the lipid bilayers of muscle mitochondria, increasing the likelihood of erythrocyte damage [28]. Additionally, excessive free radical formation and microtrauma from high-intensity exercise can elevate the inflammatory state [29].

Several studies have reported altered erythrocyte FA profiles in different groups of athletes from different sports specialties compared to control subjects and reference ranges [16,18,19,30,31]. However, few longitudinal studies have been conducted in athletes to observe changes in various FAs over a sports season. In a recent study, Peña et al. [32] confirmed changes in different FAs in a women’s soccer team over multiple seasons, a sport characterized by high levels of competitive activity and high-intensity training. Nevertheless, the periodization of loads and competitions in endurance runners differs significantly during the season. They have lower competitive density and train high volumes, with the majority at moderate–low intensity and a small percentage at high intensity, above the second ventilatory threshold (VT_2_) (1). We hypothesize that regular training will induce changes in erythrocyte FA percentages in runners, particularly during periods of increased competitive activity and high-intensity training. Therefore, this study aimed to determine the FA profile in erythrocytes in a group of high-level endurance runners at different points throughout a sports season and observe changes resulting from modifications in weekly training volumes and intensities during different periods.

## 2. Materials and Methods

### 2.1. Participants

The study was carried out on 21 high-level male endurance runners (23 ± 3 years old; height: 1.76 ± 0.04 m; 65.50 ± 7.30 kg) who resided within the same geographical region. They had personal bests between 3:37.79 to 4:08.24 in the 1500 m and from 13:11.01 to 15:10.35 in the 5000 m. Also, runners exhibited values over 65 mL/kg/min of VO_2_ max [33]. The study sample was the same as in previous research [34,35]. Four samplings were carried out every 3 months during a sports season.

All the subjects were healthy and had to meet the following inclusion criteria: (i) to be men; (ii) to compete at the national and international level in endurance races; (iii) to have at least 5 years of competing experience; (iv) to carry out more than 5 weekly training sessions and 70 km per week during the season; (v) not to change nutritional habits during the research; (vi) no weight changes >3% during the season. The exclusion criteria were: (i) not completing 25% of the training sessions due to injury or illness and (ii) ingesting ω-3 supplements during the research or any supplements in the three weeks previous to the samples’ collection.

Informed written consent was secured from all participants. The study protocol received approval from the Ethics Committee of the University of Extremadura (52/2012) and was conducted in accordance with the principles outlined in the 1975 Helsinki Declaration, as revised by the World Medical Assembly in Fortaleza, Brazil, in 2013, for research involving human subjects.

### 2.2. Study Design

Four samples were taken from the runners during a sports season, one every 3 months. The initial evaluation (Initial) was carried out in the first week of October, after an adaptation period of about 2–3 weeks when the runners only performed sessions below VT_2_, since they had had a rest period of between 2–3 weeks without training at the end of the previous season. In the first week of January, the second evaluation (3 months) was performed after completing the first preparatory period from October to December. The third evaluation (6 months) was carried out the first week of April, after finishing the first competitive period when the athletes carried out cross-country competitions. Finally, in the first week of July, the fourth evaluation (9 months) was performed, when the runners had completed a second but smaller preparatory period between April and May and were in the second competitive period that took place between June and July when the runners competed in a track-field competition. After the first competitive period, the runners had 2 weeks of transition, with less volume and without high intensity sessions.

### 2.3. Nutritional Assessment

The nutritional evaluation was carried out following the same methodology used in previous studies by our research group [36]. All participants completed a 3-day dietary record using the provided nutritional questionnaire, which included one weekend day and two weekdays. Each participant meticulously weighed and recorded the amount of each food item consumed in grams. The athletes’ dietary intake was analyzed using a comprehensive food composition table [37]. Table 1 shows the athletes’ nutritional intake during the season. The athletes adhered to a diet formulated according to established energy and macronutrient guidelines [38].

### 2.4. Anthropometric and Ergoespirometric Measures

Table 2 shows the anthropometric parameters and ergoespirometric characteristics in the runners during the season. Anthropometric measurements were performed on athletes with an empty stomach and always at the same time, between 9 and 10 a.m. Body weight (Seca 769, Hamburg, Germany) and height (Seca 220, Hamburg, Germany) were measured to the nearest 0.1 kg and 0.1 cm, respectively, in accordance with the manufacturer’s guidelines. Six skinfolds (subscapular, triceps, supraspinal, abdominal, thigh, and calf) were assessed using a skinfold caliper (Holtain, Crosswell, UK) with a precision of 0.1 mm. Each skinfold measurement was taken three times, with the average value recorded in millimeters. All measurements were conducted by an experienced and certified tester following the International Society for the Advancement of Kinanthropometry (ISAK) protocol. Fat and muscle weight were calculated following the instructions of ISAK [39].

After recording anthropometric measurements, the runners carried out an incremental treadmill test to exhaustion (Powerjog, Birmingham, UK) to assess their ergospirometric parameters and performance. This test was conducted using an ergospirometer system with a gas analyzer (Metamax, Cortex Biophysik, Leipzig, Germany). Additionally, a heart rate monitor (Vantage M, Polar, Finland) was employed to measure the maximal heart rate.

After a 10 min warm-up, the runners began the test at a speed of 10 km/h, with the speed increasing by 1 km/h every 400 m until they reached voluntary exhaustion. VO_2_ max was determined based on the following criteria: a plateau in oxygen uptake (VO_2_), an increase in carbon dioxide (CO_2_) elimination, an increase in ventilatory volume (VE) due to the increased test velocity, and a respiratory exchange ratio (RER) exceeding 1 [40]. The aerobic threshold VT_1_ and VT_2_ were identified according to the three-phase model for monitoring training and internal load [41].

### 2.5. Training Characteristics

A GPS package equipped with a heart rate monitor (Polar Electro, Polar Vantage M. Kempele, Finland) was used to track training loads throughout the season. Runners quantified the internal load through total time in each training zone [1]. Table 3 shows the training characteristics of the runners. The athletes engaged in two to three resistance training sessions per week throughout the athletics season. Generally, the exercise volume was moderate to high (2–4 sets of 4–16 repetitions) while the intensity was low–moderate (30–70% of 1RM).

### 2.6. Sample Collection

Following the anthropometric assessment of the athletes, venous blood samples were drawn from the antecubital vein into 10 mL tubes containing Ethylene Diamine Tetra Acetic Acid (EDTA). These samples were immediately centrifuged at 3000 rpm for 10 min. Subsequently, erythrocytes were washed thrice with 0.9% sodium chloride (NaCl). The erythrocytes were then transferred to sterile tubes and stored at −80 °C until analysis.

### 2.7. Analytical Determination

FA concentrations were determined in erythrocytes using the technique described by Lepage and Roy [42]. A gas chromatograph HP-5890 Series II equipped with a Flame Ionisation Detector (FID) was utilized. The analysis was conducted on a BP × 70 capillary column (50 m × 0.22 mm I.D., 0.25 µm film thickness, Cromlab, Barcelona, Spain). The initial oven temperature was set to 170 °C and maintained for 15 min. It was then increased to 190 °C at a rate of 3 °C/min and held for 15 min, followed by an increase to 245 °C at 3 °C/min, with the final temperature held for 30 min. Helium (He) was used as the carrier gas at a flow rate of 1.0 mL/min. The injector operated in splitless mode at 300 °C, with a purge flow of 6 mL/min applied 0.5 min post-injection. The FID was maintained at 250 °C.

FA identification was achieved by comparing the retention times of the FA methyl esters with those of known FA standards under identical chromatographic conditions, using retention parameters relative to an internal standard. Heptadecanoic acid was selected as the internal standard due to its similarity to the analyses and its distinct chromatographic position, which did not overlap with other sample peaks. FA concentrations were expressed as a percentage of total FAs (relative %), with over 97% of GC peaks being accurately identified using appropriate standards.

### 2.8. Lipid Profile of the Erythrocyte Membranes

Ten FAs were selected for analysis: for Saturated Fatty Acids (SFAs), palmitic acid (PA) and stearic acid (SA); for Monounsaturated Fatty Acids (MUFAs), oleic acid (OA); for ω-3 PUFAs, alpha-linolenic acid (ALA), docosapentaenoic acid (DPA), eicosapentaenoic acid (EPA), and docosahexaenoic acid (DHA); for ω-6 PUFAs, linoleic acid (LA), calendic acid (CA), and AA. Based on these FAs, various indices were calculated: the Saturation Index (SI), calculated as the ratio of % SFAs to % MUFAs, relating to membrane fluidity; the ω-3 index (ω-3 IND) as the sum of DHA and EPA; the inflammatory risk index as the ratio of % ω-6 to % ω-3. Additionally, enzymatic indices for elongase (SA/PA), ω-3 desaturase (DHA/DPA), and delta-9 desaturase (OA/SA) activities were calculated. Optimal value ranges for each of the ten FAs were derived from existing literature [24].

### 2.9. Statistical Analysis

The statistical analysis was conducted using IBM SPSS Statistics software version 21.0 (IBM Co., Armonk, NY, USA). The results are presented as x ± s, where x represents the mean values and s denotes the standard deviation. Prior to analysis, all variables were tested for normality using Kolmogorov–Smirnov tests. Data were analyzed via repeated measures analysis of variance (ANOVA) with the Bonferroni post hoc test for moment/period as the categorical variable. The equality of variances between differences was evaluated using Mauchly’s test of sphericity. When sphericity was violated, Greenhouse–Geisser corrected *p*-values were applied. Simple linear regression analysis was performed to investigate associations between FAs, the ω-3 index, the ω-6/ω-3 ratio, and kilometers trained per week. A *p*-value of less than 0.05 was considered statistically significant.

## 3. Results

The profile of fatty acids in erythrocytes in the runners throughout the sports season is shown in Table 4. Within the SFAs, we can observe a very significant increase (*p* < 0.01) in PA at 3 and 9 months and significant (*p* < 0.05) at 6 months compared to the beginning of the season. A significant decrease also occurred between 3 and 6 months. In relation to SA, we found an increase (*p* < 0.05) at 3 months compared to the beginning and a decrease at 6 months compared to 3 months.

In the ω-3 PUFA family, an increase in ALA was observed after 6 months of training compared to the initial intake. In DHA, we found a very significant decrease (*p* < 0.01) at 6 and 9 months of training. No significant changes were found in EPA. In DPA, we found an increase (*p* < 0.01) at 3 months compared to the beginning and a decrease at 9 months, very significant (*p* < 0.01) compared to 3 months and significant (*p* < 0.05) in relation to 6 months of training. Finally, an increase (*p* < 0.05) in the ω-3 IND was obtained at 9 months compared to 6 months of training.

Regarding the ω-6 family, an increase (*p* < 0.05) was obtained in LA at 9 months compared to the beginning of the season and in CA between the training period from 3 to 6 months. In AA, a very significant decrease (*p* < 0.01) was observed at 3 and 6 months compared to the beginning and an increase (*p* < 0.05) at 9 months compared to the values reported at 3 and 6 months of training.

Regarding the only MUFA studied, OA reported a significant increase at 3, 6, and 9 months of the study compared to the beginning of the season.

No significant changes were found in the ω6/ω3 ratio.

The evolution of the total FA, the saturation index, and the elongase and desaturase enzymes 5 and 9 throughout the season are shown in Table 5. A significant increase (*p* < 0.05) in the total SFAs was observed at 3 and 9 months after the initial intake. In relation to total PUFAs, a decrease (*p* < 0.05) was reported at 3 and 6 months compared to the beginning of the season and an increase (*p* < 0.05) at 9 months compared to 3 months. In total MUFA, a significant increase (*p* < 0.05) was obtained during all evaluations carried out compared to the beginning. Regarding desaturase enzymes, a decrease (*p* < 0.05) was reported at 3 and 6 months compared to the beginning in ω-3 desaturase, followed by an increase at 9 months compared to 3 months. There were no significant changes in elongase or desaturase 9 in our runners.

Simple linear regressions between different FAs with km/week trained, km/week < VT_2_, Km/week > VT_2_, km/period, anthropometric parameters, and VO_2_ max were performed. Table 6 shows only those correlations that were significant. A positive relationship was observed (r = 0.243; *p* = 0.026) between PA and the total volume of km trained weekly. Furthermore, we found a positive relationship between PA (r = 0.261; *p* = 0.016), OA (r = 0.367; *p* = 0.001) and DPA (r = 0.258; *p* = 0.018) and a negative relationship between AA (r= −0.460; *p* = 0.000) and DHA (r = −0.302; *p* = 0.003) with the km trained above VT_2_. There was also a positive relationship between PA (r = 0.374; *p* = 0.000) and DPA (r = 0.260; *p* = 0.017) and a negative relationship between AA (r = −0.450; *p* = 0.000) and DHA (r = −0.293; *p* = 0.007) with the km accumulated by periods. A positive relationship (r = 0.287; *p* = 0.008) was reported between total SFA and the km accumulated in the period and a negative relationship between total PUFA and the km run at high intensity (r = −0.415; *p* = 0.002) and the km accumulated for each period (r = −0.359; *p* = 0.001). Finally, we found a negative relation between EPA with fat mass (r = −0.226; *p* = 0.038) and SA with VO_2_ max (r = −0.226; *p* = 0.038).

## 4. Discussion

The objective of the present longitudinal study was to determine the FA profile in the erythrocyte membranes of high-level endurance runners and to observe changes during a sports season in relation to their training. To the best our knowledge, this is the first study to observe changes at four different time points in the erythrocyte profile in a group of high-level endurance runners. Runners modify their training loads throughout the season to achieve optimal performance during competitive periods. Training throughout a sports season induces changes and adaptations in athletes with the aim of improving their performance [43].

Baseline concentrations of different FAs in erythrocytes at the four sampling points carried out within the sports season showed significant changes. FAs are the main component of erythrocyte membranes and their composition is considered a dynamic system [22]. Dietary intake and the type of physical exercise are two factors that significantly and directly influence their composition and regulation [44]. Therefore, changes in training volumes or high-intensity work during training sessions and competitions affect the erythrocyte membrane [30]. In general, our study shows an increase in SFAs and a decrease in PUFAs throughout the season.

Regarding SFAs, an increase in PA at 3 months of training and SA during the season was observed. Both are close to the upper limit of reference values. Various studies have reported higher percentages of SFAs in different groups of athletes than in non-athlete populations [19,44,45]. Aerobic exercise increases the mobilization of FAs from adipose tissue and lipid transport between organs and tissues and improves insulin sensitivity and mitochondrial optimization, promoting increased FA oxidation as an energy substrate [44]. Ney et al. [46] reported that chronic aerobic training in a group of swimmers increased the proportion of PUFAs in adipose tissue and led to a higher mitochondrial b-oxidation ratio of PUFAs compared to SFAs in skeletal muscle tissue and myocardium, as the body has a higher affinity for the oxidation of PUFAs than SFAs. As a result of increased PUFA oxidation as an energy substrate, an increase in SFAs occurred in the erythrocyte membrane of runners [30].

Additionally, cardioprotective effects of SA have been documented [47]. Thomas et al. [48] found, in their study, that SA concentrations positively correlate with the number of kilometers covered. Our runners’ results confirm this, where an increase in SA was observed throughout the season but was more pronounced in the first period, where runners covered the highest weekly volume of the entire season at a lower intensity. In our correlation study, runners had a positive relationship between accumulated kilometers over the period and total SFA. We found an interesting negative association between SA and VO_2_ max. SA may contribute to inflammation and endothelial dysfunction [49]; therefore, lower SA concentrations in erythrocytes could improve membrane permeability and VO_2_ max in runners.

Regarding the studied MUFA, an increase in OA throughout the season was observed compared to the initial intake. The values found are similar to those of other groups of athletes [19,30,44]. High OA percentages have a beneficial and cardioprotective effect as they reduce the risk of coronary heart disease [50]. Deficiencies in PUFAs induce a strong compensation in different tissues, increasing OA concentrations [51]. Runners increase PUFA oxidation as a result of endurance training, leading to reduced concentrations in the body, compensated by increased OA. It has also been reported that OA is more stable than PUFAs and can partially replace them in the erythrocyte membrane to provide stability, protecting against oxidative stress [51]. It is widely known that physical exercise increases the production of free radicals, which can induce oxidative stress if not balanced by antioxidant systems [52]. Therefore, AO percentages in erythrocytes increase in runners due to the reduction in PUFA concentrations, serving as an adaptation to stabilize the erythrocyte membrane and protect it from potential oxidative stress resulting from training throughout the season.

Within PUFAs, we have the ω-3 and ω-6 families, which have opposing effects on human metabolism and compete for the same metabolic enzymes. An imbalance in the ω-6/ω-3 ratio can cause alterations in the composition of the cell membrane, affecting its fluidity and function [53]. The ω-6/ω-3 ratio of our runners was within reference values throughout the season, with no significant changes.

Within the ω-6 family, changes in different FAs were reported throughout the season. In AA, elevated average values were observed compared to standard values in the control population but similar to other athletes [30,44]. High AA concentrations allow an increase in PUFAs in erythrocyte membranes, which is related to increased membrane fluidity, having a positive effect on athletes by improving oxygen diffusion [7]. On the other hand, AA is established as a precursor of pro-inflammatory eicosanoids such as thromboxanes, leukotrienes, and prostaglandins, which, in excess, can contribute to an increase in the inflammatory state [54]. However, a recent review by Mitchell and Kirkby [55] reported a positive correlation between high AA values and decreased blood pressure. Prostacyclin is an eicosanoid derived from AA through the action of the enzyme cyclooxygenase isoform 2 (COX-2), acting as a potent platelet aggregation inhibitor and vasodilator [55]. Therefore, elevated AA values are a possible adaptation in runners that facilitates their performance as a result of training.

During the season, runners reported a decrease in AA at 3 and 6 months of training compared to the initial evaluation. Andersson et al. [56] reported a reduction in AA values in a group of young individuals after a period of low-intensity endurance training. This is consistent with the results observed in our runners, as during the first 6 months, they accumulated a higher volume of aerobic work. As mentioned earlier, during aerobic exercise, runners increase PUFA oxidation relative to SFAs due to their higher affinity, leading to a reduction of these in erythrocytes.

Subsequently, AA concentrations increased at 9 months compared to 3 months. In high-intensity activities, the body primarily obtains energy through glycolysis [57], reducing PUFA oxidation. Additionally, this type of high-intensity exercise increases the generation of superoxide radicals in the lipid bilayers of erythrocytes [58]. Excessive ROS production induces a systemic inflammatory response, increasing the synthesis of proinflammatory cytokines that must be balanced by anti-inflammatory systems [29]. The increase in AA in runners would be a consequence of a higher cellular demand for pro-inflammatory eicosanoids, leukotrienes B4, thromboxanes A2, and prostaglandins E2, which regulate the synthesis and release of proinflammatory interleukins such as TNF-alpha, IL1, IL6, and IL10 due to high-intensity training. Therefore, the increase in AA at 9 months would be a consequence of the accumulation of loads during the season, especially due to the increase in the intensity of training sessions and competitions during this period.

EPA and DHA are the most studied ω-3 PUFAs in athletes. They are fundamental components of lipid bilayers in cell membranes, increasing their structural integrity [59]. EPA and DHA compete with AA for being incorporated into cell membranes, helping to reduce chronic inflammation, acting acutely through inflammation mediators such as prostaglandins, leukotrienes, lipoxins, resolvins, and protectins [11]. They can be synthesized through the essential fatty acid ALA, although their synthesis is very limited in humans [60]. EPA concentrations in runners are higher than reference ranges in non-athlete populations [30,44]. Although an increase in their concentrations was observed throughout the season, these changes did not become significant. It has been reported that EPA competes with AA for cyclooxygenase (COX) enzyme metabolism; therefore, its elevated values during the season would aim to provide protection to erythrocytes by balancing the production of pro-inflammatory eicosanoids and cytokines due to training [12]. A negative association between EPA and fat mass was found in our study. Previous studies reported that positive effects of PUFAs on body fat mass loss may be associated with a modulation of lipid metabolism and muscle anabolism in obese subjects [61,62]. More studies in runners are necessary to investigate this association. Regarding DHA, concentrations are lower than optimal values from 3 months of training onwards. However, values are similar or even higher than those reported in other male athlete groups from different sports such as rugby (3.77 ± 0.94), basketball (3.05 ± 0.85), soccer (3.10 ± 0.73), and boxing (3.04 ± 0.99) [19,30,62]. Additionally, a significant decrease was observed at 6 and 9 months. It has been reported that DHA is more effective than EPA in modulating inflammation and lipid markers [63]. Martorell et al. [22] reported that DHA supplementation in high-level athletes during 8 weeks of training reduced lipid peroxidation and erythrocyte damage compared to a control group as a result of improved superoxide dismutase. In our runners, DHA concentrations decreased in periods of increased high-intensity sessions, as well as the number of competitions, which could be a consequence of increased antioxidant activity in erythrocytes to counteract the excess ROS induced by this type of activity.

ALA concentrations are higher than those reported in other athlete groups [64]. ALA is obtained from vegetables and is a precursor to DHA and EPA, with its conversion being very limited in humans, although there is great variability, even in subjects following the same diet [11]. In our runners, a very significant increase occurred at 6 months, when athletes had completed the first competitive period. We do not have an explanation for this modification. We hypothesize that a higher inflammatory state resulting from high-intensity training and competitions would increase the synthesis of pro-inflammatory eicosanoids from AA and lower the synthesis of anti-inflammatory eicosanoids from EPA, as both compete for COX and lipoxygenase.

Regarding DPA, values are higher than those reported in other athletes [44,65]. DPA is one of the lesser-known and -studied ω-3. In humans, circulating DPA levels appear not to be related to nutritional intake, suggesting an endogenous origin through EPA elongation [65]. Dalli et al. [66] reported that DPA plays a fundamental role in the resolution of inflammatory processes and the regulation of the immune system. Recently, Ervik et al. [67] demonstrated that one of the specialized pro-resolving mediators (SPMs), such as the DPA-derived pro-resolvin RvD5 ω-3 DPA, can be biosynthesized from DPA when its values are elevated, acting as a potent positive regulator in the elimination of fungi and bacteria, as well as decreasing inflammation markers. This would justify the changes in this ω-3 in our athletes, where an increase was observed at 3 months, indicating a higher anti-inflammatory capacity as a positive adaptation to the higher volume of moderate-low-intensity training in this period, followed by decreases at 6 and 9 months of training compared to 3 months. As mentioned earlier, high-intensity training would reduce its concentrations throughout the season as a result of increased ROS production and inflammation in runners. In this respect, our runners reported a negative relationship between total PUFAs and kilometers covered at high intensity > VT_2_.

A limitation of the study is that the nutritional intake of the runners during the season is based on nutritional surveys from the week before sample collection, and erythrocytes have a lifespan of 120 days. Nutritional intake significantly affects the composition of the erythrocyte membrane. In this context, previous studies evaluating the lipid profile in erythrocytes across different groups of athletes also conducted nutritional assessments using different questionnaires [8,31,44], with the aim of detecting significant changes in the athletes’ eating habits. Other studies did not report nutritional evaluations [18,19,20]. Conducting a reliable nutritional intake assessment is challenging due to the multitude of factors affecting the nutritional composition of foods, such as the soil origin of fruits and vegetables, the diet provided to animal-based foods, and their cooking processes [68]. In our research, runners were encouraged to maintain a stable weekly nutritional intake during the season, according to established energy and macronutrient guidelines, ref. [38], making adjustments based on kilometers covered and intensity. Other limitation found is that the study was conducted only in men. A future line of research is to conduct a study on female endurance runners and observe the potential differences compared to high-intensity sports such as soccer [32]. Finally, we had a small sample due to the limited number of subjects with such specific characteristics. This leads to limited information to properly justify the results. Therefore, more studies are needed to confirm these results.

## 5. Conclusions

Changes in training, both in volume and intensity, modify the FA profile in the erythrocyte membranes of high-level endurance runners throughout a season. Training periods with higher volumes of aerobic kilometers lead to increased concentrations of SFAs and MUFAs, coupled with a reduction in concentrations of PUFAs such as AA and DHA. A deficit in ω-3 concentrations in high-level endurance runners has a negative effect on resolution of inflammation and their long-term health. Therefore, endurance runners should pay special attention to the intake of PUFAs ω-3 in their diet or consider supplementation during training periods to avoid deficiency.

## Figures and Tables

**Table 1 nutrients-16-01895-t001:** Nutritional intake of lipids and fatty acids during the season.

Parameters	INITIAL	3 MONTHS	6 MONTHS	9 MONTHS
Lipids (g/kg/d)	1.82 ± 0.85	1.42 ± 0.57	1.40 ± 0.44	1.77 ± 0.86
Saturated fatty acids (g/day)	37.45 ± 21.14	31.45 ± 11.14	33.45 ± 14.54	36.15 ± 19.17
Monounsaturated fatty acids (g/day)	52.27 ± 25.24	40.76 ± 17.24	38.03 ± 12.24	43.45 ± 22.18
Polyunsaturated fatty acids (g/day)	11.97 ± 5.35	12.11 ± 4.69	11.56 ± 3.93	13.08 ± 8.58
ω-6 (g/day)	9.28 ± 4.82	9.08 ± 4.84	9.47 ± 4.00	10.58 ± 6.34
ω-3 (g/day)	1.08 ± 0.44	0.98 ± 0.53	1.03 ± 0.44	1.33 ± 1.11

**Table 2 nutrients-16-01895-t002:** Anthropometric and ergoespirometrics characteristics in the runners during the season.

Parameters	INITIAL	3 MONTHS	6 MONTHS	9 MONTHS
VO_2_ max (mL/kg/min)	68.30 ± 4.45	67.82 ± 8.23	68.80 ± 6.73	68.62 ± 7.37
VT_2_ (%VO_2_ max)	90.84 ± 2.68	92.56 ± 3.27	91.04 ± 3.44	90.71 ± 2.05
vVT_2_ (Km/h)	19.37 ± 0.90	20.08 ± 0.80 **	19.76 ± 1.10	19.48 ± 1.40
Maximum heart rate (bpm)	190 ± 9	192 ± 7	194 ± 9	193 ± 7
Weight (kg)	65.50 ± 7.30	65.45 ± 7.36	64.67 ± 7.03 *	64.80 ± 7.34 *
Fat mass (kg)	5.59 ± 1.23	5.42 ± 1.07	5.24 ± 0.83 *	5.24 ± 0.96 *
Muscle mass (kg)	32.19 ± 4.00	32.36 ± 4.01	31.83 ± 3.93	31.88 ± 4.12 *

VO_2_ max: maximal oxygen consumption; VT_2_: second ventilatory threshold; vVT_2_: running speed at second ventilatory threshold; bpm: beats per minute. * *p* < 0.05 initial vs. 3, 6, 9 months. ** *p* < 0.05 initial vs. 3, 6, 9 months.

**Table 3 nutrients-16-01895-t003:** Training characteristics in the runners during the season.

	INITIAL	3 MONTHS	6 MONTHS	9 MONTHS
Training (km/week)	44.32 ± 8.16	114.78 ± 18.26	101.11 ± 15.54	80.90 ± 13.36
<VT_2_ (km/week)	44.32 ± 8.16	91.83 ± 14.61	75.83 ± 11.66	69.62 ± 11.36
>VT_2_ (km/week)	-	22.96 ± 3.65	25.28 ± 3.89	12.29 ± 2.01

VT_2_: anaerobic threshold.

**Table 4 nutrients-16-01895-t004:** Evolution on FAs during the season.

Parameters	RANGES	INITIAL	3 MONTHS	6 MONTHS	9 MONTHS
Palmitic Acid	17–27	22. 52 ± 1.22	25.15 ± 2.40 **	23.35 ± 0.96 *^$^	24.94 ± 2.19 **^##^
Stearic Acid	13–20	19.46 ± 0.93	20.51 ± 1.71 *	19.67 ± 0.97 ^$^	20.52 ± 2.36
Oleic Acid	9–18	16.73 ± 1.01	17.32 ± 1.34 *	17.35 ± 1.10 *	17.42 ± 1.27 *
Linoleic Acid	9–16	11.81 ± 1.19	11.83 ± 1.55	11.69 ± 0.93	12.75 ± 2.40 *
Calendic Acid	-	0.60 ± 0.22	0.58 ± 0.17	0.77 ± 0.33 ^$^	0.64 ± 0.43
Alpha Linoleic Acid	-	0.88 ± 0.38	0.83 ± 0.33	1.59 ± 0.24 **^$$^	0.72 ± 0.31 ^++^
Arachidonic Acid	13–17	20.22 ± 2.42	17.41 ± 2.27 **	17.77 ± 1.60 **	19.10 ± 2.44 ^+^
Eicosapentaenoic Acid	0.5–9	0.98 ± 0.33	1.20 ± 0.52	1.15 ± 0.50	1.36 ± 0.99
Docosapentaenoic Acid	-	1.69 ± 0.53	2.45 ± 0.90 **	1.88 ± 0.51 ^$^	1.63 ± 0.60 ^+^
Docosahexaenoic Acid	5–7	5.07 ± 1.04	4.36 ± 1.89	3.96 ± 0.93 **	4.46 ± 1.12 **
Index ω-3	>8%	6.06 ± 1.09	5.56 ± 2.16	5.11 ± 1.02	5.82 ± 1.22 ^#^
ω-6/ω-3 ratio	3.5–5.5	4.24 ± 0.80	3.95 ± 1.00	4.38 ± 1.16	4.40 ± 0.88

* *p* < 0.05 between 0 vs. 3/6/9 months; ** *p* < 0.01 between 0 vs. 3/6/9 months; $ *p* < 0.05 between 3 vs. 6/9 months; $$ *p* < 0.01 between 3 vs. 6/9 months; # *p* < 0.05 between 6 months vs. 9 months; ## *p* < 0.01 between 6 months vs. 9 months; + *p* < 0.05 between 3 months vs. 9 months; ++ *p* < 0.01 between 3 months vs. 9 months.

**Table 5 nutrients-16-01895-t005:** Changes in total FAs and enzymes during the season.

Parameters	RANGES	INITIAL	3 MONTHS	6 MONTHS	9 MONTHS
Total saturated fatty acids	34–45	41.99 ± 1.94	45.67 ± 3.74 *	43.03 ± 1.74	45.47 ± 3.85 *
Total polyunsaturated fatty acids	30–43	39.79 ± 2.18	37.27 ± 2.81 *	36.48 ± 1.64 *	39.31 ± 4.47 ^$#^
Total monounsaturated fatty acids	15–23	16.74 ± 1.02	17.33 ± 1.34 *	17.35 ± 1.11 *	17.42 ± 1.27 *
Saturation index	1.7–2	2.52 ± 0.17	2.60 ± 0.29	2.49 ± 0.13	3.01 ± 0.95
Desaturase 9	-	0.86 ± 0.06	0.85 ± 0.11	0.88 ± 0.05	0.86 ± 0.08
Desaturase ω-3	-	3.32 ± 1.32	1.91 ± 0.82 *	2.30 ± 1.15 *	3.13 ± 1.41 ^#^
Elongase	-	0.86 ± 0.76	0.82 ± 0.71	0.84 ± 1.01	0.82 ± 1.08

* *p* < 0.05 between 0 vs. 3/6/9 months; $ *p* < 0.05 between 3 vs. 6/9 months; # *p* < 0.05 between 6 months vs. 9 months.

**Table 6 nutrients-16-01895-t006:** Simple linear regression between erythrocytes FAs and training performed, anthropometric parameters, and VO_2_ max.

Parameters	Km/week		Km/week >VT_2_		Km/period		Fat mass		VO_2_ max (mL/kg/min)	
	r	*p*	r	*p*	r	*p*	r	*p*	r	*p*
Palmitic Acid.	0.243	0.026	0.261	0.016	0.374	0.000				
Stearic Acid.									−0.227	0.038
Alpha Linoleic Acid	-	-	0.367	0.001	-	-				
Arachidonic Acid.	-	-	−0.460	0.000	−0.450	0.000				
Eicosapentaenoic Acid							−0.226	0.038		
Docosapentaenoic Acid	-	-	0.258	0.018	0.260	0.017				
Docosahexaenoic Acid	-	-	−0.302	0.005	−0.293	0.007				
Total saturated fatty acids	-	-	-	-	0.287	0.008				
Total polyunsaturated fatty acids	-	-	−0.415	0.002	−0.359	0.001				

VO_2_ max: maximal oxygen consumption; VT_2_: second ventilatory threshold.

## Data Availability

The raw data supporting the conclusions of this article will be made available by the authors on request.

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
