# Peer review of "Changes in the Fatty Acid Profile in Erythrocytes in High-Level Endurance Runners during a Sports Season"

_nutrients, 2024, doi:10.3390/nu16121895_

Round 1
Reviewer 1 Report
Comments and Suggestions for Authors
Fatty acids (FAs) are an essential component of the erythrocyte membrane, and nutrition, and physical exercise are two variables that affect its structure and function. So, the aim of this study was to evaluate the erythrocyte profile in a group of high-level endurance runners, as well as the changes in different FAs throughout a sports season in relation to the training performed. As a results, authors in the present study suggested that training modifies the erythrocyte FA profile in high-level endurance runners, reducing the concentrations of polyunsaturated fatty acids (PUFAs) such as DHA and AA, and increasing the concentrations of saturated fatty acids (SFAs) such as the SA and the PA.
This manuscript is written well. A few points will need to revise.
1)Please provide more details about the subject selection criteria.
2)Are there any other limitations to this study?
Comments on the Quality of English LanguageSpell check will need before publishing.
Author Response
Dear editor:
I am writing to reply point-by-point to the reviewers' comments and the editorial office's requests. Firstly, I am very grateful for the considerations of the comments done regarding the first version of the paper. The revised article has improved in some points. The changes have been highlighted in red colour in the revised manuscript.

Reviewer 2 Report
Comments and Suggestions for Authors
Dear Corresponding Author,
Thank you for submitting your paper, and congratulations on your work. Your study aims to examine the variations in the fatty acid profile of high-level endurance runners during a sporting season. The research offers interesting contributions regarding the adaptation of erythrocyte fatty acids to the training load, with significant results that could have practical implications for endurance athletes. However, there are significant criticisms that I would like to be clarified.
General Comments: The manuscript presents some areas of weakness, including the limitation of the study sample to only men and the reliance on nutritional questionnaires for the assessment of dietary intake, which could affect the validity of the results. Further insight into the control of nutritional variables and greater diversification of the sample would be useful.
Specific Comments:
- Line 88: You define the athletes as "High Level," but to use this definition, it would be necessary to understand the ranking score of the World Athletics Rankings or similar.
- Lines 90-95: Clarify the exclusion criteria related to nutritional habits and the use of FA supplements.
- Table 2: Provide more details and references on the anthropometric assessment methods.
- Line 132: You state that you measured the Training Load with GPS. It seems you measured the kilometers covered by the athletes but not the actual individualized workload (and internal load) for each athlete.
- Line 138: By what method or test did you analyze the "anaerobic threshold"?
- Line 272: When referring to "elite athletes" who should have a very well-periodized training plan, it seems extremely unlikely to define it as "aerobic exercise." Based on which physiological assessment?
- Lines 321-325: Add further discussions on the limitations of nutritional assessment based on questionnaires and their impact on the fatty acid profile.
- Line 328: The conclusions are extremely simplistic, but the work has significant limitations regarding the training load that do not seem to have been appropriately considered.
In this form, it is not considered acceptable for publication, but we give the authors the opportunity to reflect on appropriate modifications and additions.
Best regards
Author Response

(The authors gave the same response as above.)

Reviewer 3 Report
Comments and Suggestions for Authors
The authors provide an interesting study about the fatty acid profile in endurance runners during a season. The result conclusively suggest that supplementation of omega fatty acids might be justified in this group of consumers.
I think the paper is conclusively written and I have only some minor remarks:
Line 25: The "runners reported" sounds strange. Please revise sentence.
Line 3: delete fullstop after title and capitalize title
Line 4: check template for correct authors list formating
Line 7: provide affiliations with postal address
Line 31: semicolons between keywords
Line 87 and throughout: sections should be numbered, e.g. 2.1
Line 120: I do not think that omega needs explanation
Comments on the Quality of English Language
Line 3: delete fullstop after title and capitalize title
Line 4: check template for correct authors list formating
Line 7: provide affiliations with postal address
Line 31: semicolons between keywords
Line 87 and throughout: sections should be numbered, e.g. 2.1
Line 120: I do not think that omega needs explanation
Author Response

(The authors gave the same response as above.)

Round 2
Reviewer 2 Report
Comments and Suggestions for Authors
I have carefully read the authors' responses, although some criticalities and outdated references remain with respect to the objective (for example, Skinner and McClellan's three-phase model from 1980), I believe that in this form it can contribute to research in this field.
Author Response
Dear reviewer, we provide a point-by-point response to the reviewer’s comments.
